# A Rapid Systematic Review Assessing the Effectiveness of Interventions to Promote Self-Management in Workers with Long-Term Health Conditions and Disabilities

**DOI:** 10.3390/ijerph21121714

**Published:** 2024-12-23

**Authors:** David W. Maidment, Katie Clarkson, Emma V. Shiel, Karina Nielsen, Jo Yarker, Fehmidah Munir

**Affiliations:** 1School of Sport, Exercise and Health Sciences, Loughborough University, Loughborough LE11 3TU, UK; k.clarkson@lboro.ac.uk (K.C.); f.munir@lboro.ac.uk (F.M.); 2School of Human and Health Sciences, University of Huddersfield, Huddersfield HD1 3DH, UK; emma.shiel@hud.ac.uk; 3Institute of Work Psychology, Management School, University of Sheffield, Sheffield S10 1FL, UK; 4Affinity Health at Work, London SW12 9NW, UK; jo.yarker@affinityhealthatwork.com

**Keywords:** disabilities, long-term conditions, occupational health, rapid review, self-management, workplace

## Abstract

The objective of this study was to synthesise evidence assessing the effectiveness of workplace-based interventions that promote self-management of multiple long-term conditions or disabilities, e.g., type I and II diabetes, asthma, musculoskeletal injury/disorder, cancer, and mental ill-health. A prospectively registered rapid systematic review was conducted. Both academic and grey literature databases were searched for papers published within the last 10 years, reflecting the most contemporary legislation and policy. The outcomes included work productivity, work engagement, self-management, work ability, quality of life, psychological wellbeing, workplace fatigue, job satisfaction, work-based attendance, work self-efficacy, and condition-specific health status. Five randomised controlled trials were included, and all assessed a chronic disease self-management programme consisting of six weekly facilitator-led group sessions. Due to the small number of studies and the variability in outcome measures employed, meta-analyses were not feasible. However, a narrative synthesis indicated that work engagement, self-management and wellbeing improved in the intervention compared to the control groups. All of the other outcomes showed mixed results. The findings highlight the need to develop less resource-intensive workplace-based self-management interventions that cater to both workers and employer stakeholders, to examine intervention implementation processes as well as effectiveness, and to draw on a common set of outcome measures to enable comparative analysis to better inform public health policy and practice.

## 1. Introduction

Long-term conditions and disabilities can refer to any prolonged (≥12 months) health issue (e.g., mental, physical, sensory) that has a substantial impact on an individual’s ability to perform daily activities, including work. Long-term health conditions and disabilities can have a profound effect on quality of life and contribute substantially to healthcare costs worldwide. In the United Kingdom (UK), for example, there are an estimated 5.5 million workers with disabilities [1] and around 3.8 million workers with at least one work-limiting long-term health condition [2]. In the United States of America (USA), it is projected that around 75–78% of workers have a minimum of one long-term health condition [3,4]. Loss of working hours and productivity due to long-term health conditions and disabilities is estimated to cost the USA and several European nations USD 5 billion [5] and USD 13 billion [6], respectively. In the UK, long-term health conditions and disabilities cost the economy approximately GBP 100 billion (USD 74 billion) per year due to lost productivity, reduced income tax receipts, increased long-term sickness (up to 131 million days lost annually), increased caregiving, and higher welfare/healthcare costs [7].

Keeping people with a long-term health condition or disability in work is challenging. Most long-term health conditions and disabilities require ongoing self-management over an individual’s lifespan because they cannot be cured. Self-management refers to the actions taken by an individual to recognise, treat, and manage their own health [8]. Examples include monitoring and responding to symptoms, taking medication or other therapies, accessing suitable support, or making practical changes (e.g., working flexibly, adapting work tasks) to maintain health.

People living with long-term health conditions and disabilities can lead a productive working life if they can successfully integrate self-management tasks with work demands [9]. Evidence shows that workers who can use work-relevant strategies over which they have control are more likely to remain in and thrive at work [9,10]. This can be successfully achieved through the combined effort of employers, occupational health (OH) and human resources (HR) professionals/providers, healthcare services, and workers [11]. Employers, for instance, can introduce management policies and practices that offer greater flexibility in terms of work adjustments [12], as well as encouraging workers and managers to work collaboratively to solve issues together [13]. A more proactive role on the part of OH providers in supporting workers’ work-relevant self-management goals can also be beneficial [14]. Nevertheless, a growing body of research shows that individuals with long-term health conditions and disabilities experience difficulties self-managing their symptoms at work [15,16,17]. The reasons for this can be complex and multi-faceted but have been attributed to stigma, work demands, and lack of knowledge from employers concerning how to best promote self-management [16]. On this basis, there is a need to identify suitable intervention strategies that can be employed by all relevant stakeholders to promote effective self-management in the workplace.

Several systematic reviews have assessed the effectiveness of self-management interventions in individuals with long-term health conditions and/or disabilities delivered in primary or secondary healthcare settings. For example, Panagioti and colleagues [18] included 184 studies assessing self-management interventions delivered in health service settings that incorporated patient education, supported decision-making, self-monitoring, and/or social support. Overall, the authors found that such interventions provided significant but minimal impacts on health outcomes (e.g., reduced health service utilisation, improved quality of life) for individuals with cardiovascular and respiratory disorders, diabetes (type I and II), and mental ill-health. Other systematic reviews assessing self-management interventions have identified improvements in the self-efficacy to manage health for several conditions, including multiple sclerosis and stroke [19,20], as well as improvements in health behaviours, such as reduced alcohol intake for individuals with stroke [21] and improved foot care for people with type II diabetes through video-based educational interventions [22]. Nevertheless, while self-management interventions delivered in healthcare settings have been shown to be effective, they may not easily translate to a workplace context. This may be because employers lack the necessary infrastructure to promote self-management (e.g., training to improve manager knowledge and awareness) and/or self-management is not being prioritised as an organisational objective [16].

Reviews assessing workplace-based interventions have typically focused on promoting the return to work following a period of sickness or disability leave. For example, in a recent systematic review by Wilson and colleagues [23], it was found that interventions using physical activity, especially aerobic and resistance exercises, had a significant positive impact on the return to work for cancer survivors. Additionally, Dibben and colleagues [24] identified that workplace-based rehabilitation interventions can assist with an effective return to work, specifically for people living with musculoskeletal injury/disorders. In addition to the return to work, several other systematic reviews have assessed vocational and workplace-based rehabilitation interventions that aim to increase work participation more generally [25,26]. The findings across reviews suggest that, while such interventions can effectively support workers with long-term health conditions and disabilities to remain in employment and reduce absenteeism, further high-quality studies, namely, randomised controlled trials, are needed.

Nevertheless, to date, no review has been undertaken that specifically assesses interventions that aim to promote self-management of long-term health conditions and disabilities in the workplace. Therefore, this study aimed to provide an up-to-date synthesis of evidence assessing the effectiveness of workplace-based interventions that promote self-management of multiple long-term health conditions or disabilities. As this work was part of a larger project, a rapid systematic review was considered a pragmatic alternative to a conventional systematic review, due to time and resource limitations. In addition, given that the promotion of self-management in the workplace is an increasingly urgent priority amongst public health policymakers, health services, and professional organisations [7,8,27,28], a rapid review was viewed as advantageous, addressing the “time-sensitive” needs of key stakeholders, such as employers, managers and workers, so that research evidence can be more rapidly translated into practice [29]. To ensure methodological rigour, recent best practice recommendations for rapid reviews introduced by the Cochrane Rapid Reviews Methods Group [29,30,31,32,33] were adhered to. Rapid reviews have been described as a form of evidence synthesis that streamlines systematic review methods, expediting the processes so a review can be completed in a shorter period [29]. As such, this review serves to provide a timely synthesis of evidence that could be used to inform new practice and policy recommendations that stakeholders can implement more immediately to promote self-management of long-term health conditions and disabilities in the workplace.

## 2. Materials and Methods

A draft review protocol was developed in consultation with the wider research team, comprising 12 academic experts working within the areas of occupational and public health. Additionally, a project-specific patient and public involvement and engagement (PPIE) group, consisting of 10 individuals in employment with lived experience of a long-term health condition or disability (e.g., type I diabetes, hearing loss, neurodiversity, mental ill-health), reviewed the protocol. The final, agreed protocol was registered prospectively with the International Prospective Register of Systematic Reviews (PROSPERO) (registration number: CRD42023475510). The methods are reported according to the Preferred Reporting Items for Systematic reviews and Meta-Analyses (PRISMA) checklist [34], since the extension of PRISMA for rapid reviews is currently under development and is yet to be published [35].

### 2.1. Eligibility Criteria

The inclusion/exclusion criteria were specified in terms of the participants/population, intervention(s), comparator(s)/control, outcomes, and study designs (PICOS) (see also Table 1).

#### 2.1.1. Participants/Population

Adults (≥18 years) with any long-term health condition or disability who are employees in any size or type of organization (small, medium, large). There were no limits on study participants regarding gender, ethnicity, occupation, employment type (e.g., full- or part-time), or seniority.

Long-term (or chronic, persisting, recurring) health conditions were defined as any health issue that requires ongoing management over several years that cannot be cured [36]. Long-term health conditions are typically controlled with the use of medication and/or other interventions. Examples include, but are not limited to, diabetes (type I and II), cancer, and chronic/persisting pain (e.g., arthritis), as well as cardiovascular (e.g., hypertension), chronic respiratory (e.g., asthma, chronic obstructive pulmonary disease), and chronic neurological (e.g., multiple sclerosis) disorders.

Disability was defined as having a negative, long-term (>12 months) impact on an individual’s ability to perform daily activities, including limitations on working/professional life [37]. Disabilities can be a mental, sensory, or physical condition, visible or hidden, and/or recurring. Examples of disabilities include, but are not limited to, vision impairment, deafness or hearing loss/hard of hearing, mental ill-health (e.g., bipolar disorder, depression, schizophrenia, anxiety, and personality disorders), intellectual disability, acquired brain injury (ABI), autism spectrum disorder, and physical conditions (usually impacts either mobility, dexterity, or stamina).

#### 2.1.2. Intervention(s)

Any workplace-based strategy or intervention that aims to promote self-management in workers living with any long-term health condition or disability. This was an evidence-based decision supported by PPIE input since individuals with long-term health conditions and disabilities typically experience multi-morbidities [38]. Interventions addressing a range of conditions are also more likely to meet individuals’ holistic needs. Self-management is defined as any action taken by an individual with a long-term health condition or disability to recognise, treat, and manage their health. Examples include monitoring and responding to symptoms, accessing support, and making practical changes to enhance work ability (or work–health balance) as well as overall quality of life.

#### 2.1.3. Comparator(s)/Control

Could be passive (e.g., no intervention, standard or usual care) or active (e.g., another type of self-management support).

#### 2.1.4. Outcomes

The outcomes were selected in consultation with the wider research team, our project-specific PPIE group, and a recently published core outcome set for workplace-based interventions [39]. The primary outcomes were (i) work productivity (e.g., Work Productivity Scale), (ii) work engagement (e.g., Work Engagement Scale), (iii) self-management (e.g., Patient Activation Measure), and (iv) work ability (e.g., Work Ability Index). The secondary outcomes included (i) quality of life (e.g., WHO Disability Assessment Schedule), (ii) psychological wellbeing (e.g., Warwick-Edinburgh Mental Wellbeing Scale), (iii) workplace fatigue (e.g., Occupational Fatigue Exhaustion Recovery scale), (iv) job satisfaction, (v) work-based attendance (e.g., absenteeism [including sickness or disability absence], presenteeism, turnover, work exit, ill health retirement, early retirement), (vi) work self-efficacy (e.g., Return-to-Work Self-Efficacy scale), and (vii) measures of condition-specific health status, including pathological or clinical assessments.

#### 2.1.5. Study Designs

Randomised and non-randomised controlled trials, as well as before and after studies, could be included. Expert opinions, practice or procedure guidelines, case reports, case series, abstracts from conferences, and book chapters were not included.

### 2.2. Search Strategy

All of the authors discussed and devised the search strategy in consultation with an academic librarian based at Loughborough University. Once finalised, K.C. searched the following databases on 7 December 2023, which were last updated on 23 September 2024: APA PsycINFO, APA PsycARTICLES, and MEDLINE (all via EBSCO host). In addition, to enhance the comprehensiveness and reduce publication bias, the following grey literature databases were searched: Social Care Online from the Social Care Institute of Excellence (https://www.scie-socialcareonline.org.uk/search/, accessed on 7 December 2023) and ProQuest Dissertations & Theses Global. The full electronic search strategies are provided in Appendix A. The database searches were completed in one day and were restricted to current evidence published within the last 10 years to reflect the most up-to-date employment legislation and policy, as well as in the English language only due to feasibility reasons. To identify any relevant articles that may have been missed in the database searches, the following methods were also employed: hand searching the last six months of publications from key occupational health journals, the reference lists of the included studies were reviewed, and related articles were screened by shortlisted authors.

### 2.3. Study Selection

Using Covidence (https://www.covidence.org/, accessed on 7 December 2023), two reviewers (E.V.S., K.N.) independently assessed the identified references to decide on eligibility based on the title and/or abstract. The full text was retrieved for articles that met the eligibility criteria or when there was uncertainty, such as a lack of sufficient information to make a definitive decision. The eligibility of each full text article was assessed independently by two reviewers (D.W.M., E.V.S.). Contact with study authors was not needed to resolve questions concerning eligibility, and any discrepancies between assessors were resolved via discussion between all of the authors.

### 2.4. Data Collection Process

A standardised data collection form constructed in Microsoft Word was piloted, which included key study details (e.g., corresponding author’s contact details, country), participants/population, intervention(s), comparator(s)/control, outcomes, study design, and key findings. Data extraction was undertaken independently by D.W.M. and K.C. for each included record. Discrepancies were resolved through discussion between all of the authors.

### 2.5. Risk of Bias in Individual Studies

The risk of bias was independently assessed by D.W.M. and E.V.S. for each included study. For randomised controlled trials, the Cochrane Risk of Bias 2 (RoB2) tool [40] was employed, which rates studies as “high risk”, “low risk” or “some concerns” across five domains: (i) the randomisation process, (ii) deviations from intended interventions, (iii) missing outcome data, (iv) measurement of the outcome, and (v) selection of the reported result. For each domain, reviewers could select “yes”, “probably yes”, “no”, “probably no”, and “not included”. To ensure the accurate implementation of the RoB2 guidance, an Excel tool was used (https://sites.google.com/site/riskofbiastool/welcome/rob-2-0-tool/current-version-of-rob-2, accessed on 1 February 2024). For non-randomised controlled trials, including grey literature, the Risk Of Bias In Non-randomized Studies of Interventions (ROBINS-I) tool [41] was employed.

### 2.6. Data Synthesis

The included studies were assessed to determine the suitability of their data for inclusion in a meta-analysis. Meta-analyses were planned only if the studies were broadly comparable in terms of the study design, interventions and outcomes. For continuous data, if the studies used the same outcome measure, the mean differences (MDs) with 95% confidence intervals (CI) would be calculated. When different outcome measures were used, the effect sizes would be calculated as the standardised mean differences (SMDs), dividing the mean difference between conditions by the pooled standard deviation (for between-group comparisons) or by the standard deviation of the differences (for within-group comparisons). The heterogeneity in the effect sizes across studies would be assessed using the I^2^ statistic, with its significance tested using a χ^2^ test. The I^2^ statistic quantifies heterogeneity as a percentage and can be categorised as low (0–40%), medium (41–60%), or high (61–100%) [42]. However, due to the small number of included studies, meta-analysis was not feasible. As a result, the primary and secondary outcomes were analysed at the individual study level using narrative synthesis [43].

## 3. Results

In total, 463 records were identified for screening. After removing 62 duplicate publications, the titles and abstracts of 401 records were screened (Figure 1). The full texts of 43 articles that passed this screening in accordance with the pre-specified PICO inclusion/exclusion criteria (Table 1) were obtained. Thirty-eight articles did not meet the inclusion criteria and were excluded for the following reasons: wrong intervention (e.g., focused on workers with a single health condition/disability only), wrong setting (e.g., primary or secondary healthcare setting), wrong outcomes (e.g., caregiver burden), wrong population (e.g., acute injury, such as bone fracture), and wrong study design (e.g., literature review). Therefore, five studies were included in the review.

### 3.1. Characteristics of the Included Studies

#### 3.1.1. Design and Methods

All five studies were randomised controlled trials (RCTs) published in peer-reviewed journals (Appendix A). The studies evaluated the outcomes after various follow-up durations, ranging from six months to one year.

#### 3.1.2. Participants

The total number of participants in the five studies was 1208, ranging from 91 to 411 participants. The average age of the participants ranged from 46 to 48 years. Across all of the studies, a greater proportion of females took part relative to males, and all of the studies reported the ethnicity of the participants. Three studies assessed a broad range of health conditions, including diabetes (type I and II), cardiovascular diseases, chronic respiratory diseases, chronic pain, cancer, and mental ill-health [44,45,46]. Two studies did not specify the conditions experienced by the participants [47,48].

#### 3.1.3. Interventions

All of the studies were based in the USA and assessed a chronic disease self-management programme, which in some studies had been adapted for the workplace, aimed at workers with any long-term health condition or disability. This programme typically consisted of six weekly facilitator-led group sessions on improving self-management through action planning, problem solving, and social support.

#### 3.1.4. Effectiveness

The results for each outcome domain across the studies are summarised in Appendix A.

Work engagement. Three studies assessed this outcome using a range of different validated self-reported questionnaires. While one study assessed work engagement directly using the Utrecht Work Engagement Scale [45], other studies measured work engagement indirectly, such as via organisational citizenship behaviours [45] and work-related stress [46]. Across these measures, all of the studies showed larger improvements in work engagement for the intervention compared to the control group from baseline to follow-up.

Self-management. Schopp and colleagues [47] employed the Self-Rated Abilities for Health Practices scale and Health Promoting Lifestyle Profile II, finding that both outcomes improved for the intervention compared to the control group from baseline to follow-up.

Work ability. One study, using the Work Ability Index, showed a positive intervention effect on work ability from baseline to follow-up [44]. However, two further studies found no statistically significant differences between groups when work ability was assessed using a modified version of the Work Ability Index and/or the Work Limitations Questionnaire [44,46].

Psychological wellbeing. Only assessed by Haynes and colleagues [44], who showed an improvement in the intervention relative to the control group from baseline to follow-up for both job stress and burnout.

Workplace fatigue. This outcome was only assessed by Shaw and colleagues [45], via the Occupational Fatigue Exhaustion Recovery scale, who found no statistically significant differences between groups from baseline to follow-up.

Job satisfaction. Shaw and colleagues [45] assessed this outcome via self-report and turnover intention, finding no statistically significant differences between groups. In comparison, Haynes and colleagues [44] measured organisational support and affective/normative organisational commitment, showing a positive intervention effect on both outcomes.

Work-based attendance. Assessed in terms of absenteeism [44] or the number of health days [48]. For absenteeism, Haynes and colleagues [44] found no differences between groups. However, Wilson and colleagues [48] found that, relative to the controls, the participants in the intervention group reported improvements in unhealthy days.

Work self-efficacy. Only one study assessed participants’ confidence in their ability to return to work following ill-health [45], showing no statistically significant differences between groups.

Condition-specific health status. Two studies assessed this outcome, employing a range of different validated self-report measures across multiple dimensions [46,48]. The results across studies were mixed; Smith and colleagues [46] only found intervention effects (i.e., improvements) for fatigue, while Wilson and colleagues [48] found effects for both fatigue and stress.

### 3.2. Risk of Bias

As all of the included studies were randomised controlled trials, the risk of bias was assessed using the RoB2 tool [40] only. Overall, all of the studies were judged to be “low risk” of bias for the randomisation process, deviations from intended interventions, missing outcome data, and measurement of the outcome (Table 2). Two studies were judged as “some concerns” for the selection of the reported result, as insufficient information was provided concerning whether the data were analysed in accordance with a pre-specified analysis plan [44,48]. Nevertheless, these studies were still judged as “low risk” overall due to the issues not being expected to impact the study quality.

## 4. Discussion

The current review is the first to systematically synthesise existing evidence on the effectiveness of workplace-based interventions designed to promote self-management in individuals with any long-term health condition or disability. Overall, the studies assessing work engagement, self-management and wellbeing showed greater improvements for the intervention relative to the control groups [44,45,46,47]. For all of the other outcomes (i.e., work ability, workplace fatigue, job satisfaction, work-based attendance, work self-efficacy, condition-specific health status), the results were inconsistent. These findings, therefore, identify areas where further research may be needed and can be used to inform the design, evaluation, and implementation of more effective workplace interventions.

Although the interventions across all of the included studies were similar in terms of the delivery mode and duration, several factors may account for the inconsistencies in the findings. First, the studies employed different measures to assess each outcome domain, which may have contributed to the variability in the results. Second, limited information was provided in some studies in terms of the participant characteristics, including the specific long-term health conditions or disabilities participants experienced. Third, certain outcomes, such as workplace fatigue or job satisfaction, may not have been directly targeted by the intervention evaluated. These outcomes may be beyond the control of individual workers and instead depend on changes at an organisational level, such as the workplace culture, managerial support, and/or policies. Collectively, these factors not only make it challenging to make direct comparisons between studies but may also have had an impact on the intervention effectiveness.

The inconsistencies between studies may also stem from their sole focus on effectiveness, without considering the intervention process and implementation, which may have influenced the outcomes. For example, all of the studies evaluated group-based interventions delivered in-person by a trained facilitator (e.g., researcher or healthcare professional), typically comprising of one 2.5 h session per week over a 6-week period. Whether interventions delivered via alternative formats (e.g., individual-based approaches, digital or remote delivery, variations in session length) would result in comparable outcomes warrants further investigation. Additionally, all of the studies assessed interventions in a range of different organisational settings (i.e., private, public, and third/voluntary sectors) and included participants with diverse long-term health conditions and disabilities, as well as differing job roles and levels of seniority. Further research is therefore needed to examine these factors in greater depth, which could provide valuable insights into how they might impact the intervention effectiveness.

This review did not assess whether the interventions were delivered as intended, how participants engaged in the intervention activities [49], or how the intervention was perceived by participants [50,51], which may have been found to influence the workplace intervention outcomes [50]. Future studies should evaluate process and implementation issues, thus moving beyond the simplicity of whether an intervention was effective. In doing so, it may be possible to obtain a better insight into what works for whom and in which circumstances [50], which could inform future intervention development and evaluation in this area.

These findings diverge from previous reviews assessing other workplace-based interventions, such as those promoting the return to work following sickness/disability absence or work participation more generally [23,24,25,26]. While prior reviews consistently support the effectiveness of the return to work for workers with long-term health conditions and disabilities in terms of retention and absenteeism, the current review suggests that self-management support needs may differ or change pre- and/or post-absence. For example, interventions targeting self-management may require a greater focus on ongoing support, such as flexible working arrangements and adjustments, whereas interventions aimed at individuals returning to work following a period of leave may require greater emphasis on the transition back into work, including phased return and reintegration programmes.

Additionally, reviews assessing return-to-work interventions often conclude that further high-quality evidence (i.e., RCTs) in this area is needed. By comparison, most studies included in the current review were RCTs, with a low overall risk of bias. Although the limited number of studies and the heterogeneity in outcome measures prevented definitive conclusions, workplace-based self-management interventions show promise; there are several high-quality studies that could be used by researchers, professional bodies, and policy/decisionmakers to inform evidence-based decisions.

### 4.1. Future Research and Implications for Practice

The reviewed interventions exclusively targeted workers with any long-term health condition or disability. These interventions were resource-intensive to complete, typically involving weekly group-based sessions, each lasting several hours, facilitated by a trained researcher or healthcare professional. Thus, the cost-effectiveness of these interventions on a larger scale remains questionable. Further investigation is also needed for interventions delivered via alternative formats (e.g., digital or remote delivery) and/or tailored for different workplace settings, including variations in the type, size, and sector. Additionally, none of the interventions included provided guidance or support for managers or employer stakeholders. This is problematic, since existing research [12,13,16,17] highlights the crucial role of managers in supporting workers with long-term health conditions and disabilities, but they often lack the knowledge and confidence to do so [27]. A recent qualitative study further found that workplace self-management promotion is not meaningfully offered to workers due to poor knowledge, stigma, and work demands [16]. Consequently, there is a need to develop cost-effective, less-resource-intensive, scalable self-management interventions that meet the needs of workers and other employer stakeholders.

As part of a wider project, we are developing a new intervention through collaboration with our PPIE group and various stakeholders from different work–health fields. This includes the design of interactive guidance and practical tools freely available digitally (i.e., via a website or smartphone application) that can also be downloaded for use in paper form. These resources will help workers with long-term health conditions identify their needs, address them in the workplace, and seek support to improve their work–health balance. Guidance and tools are also being designed for managers and employers to enable them to confidently respond to workers’ needs. The next steps include evaluating the effectiveness of these resources on work–health outcomes and exploring successful implementation across diverse workplace contexts and healthcare settings.

### 4.2. Review Strengths and Limitations

To the best of our knowledge, this systematic rapid review is the first to synthesise the current literature on workplace interventions for self-management of any long-term health condition or disability. To uphold rigour, best practice recommendations were followed [29,30,31,32,33], including duplicate screening at all stages by independent reviewers and consultations with a relevant PPIE group when developing the protocol and synthesising the results. However, and in accordance with these recommendations [29,30,31,32,33], a small number of information sources (i.e., APA PsycINFO, APA PsycARTICLES, MEDLINE, Social Care Online from the Social Care Institute of Excellence, ProQuest Dissertation & Theses Global) were selected that were most likely to retrieve the relevant literature, and the papers were restricted to those available in the English language and published within specific sources in the last 10 years. On this basis, it could be argued that other potentially relevant publications that fall within the scope of our review were missed, which may also explain the scarcity of included studies. In addition, restricting the review to English language publications may introduce language or cultural biases, limiting the generalisability of the findings to non-English speaking and/or low- and middle-income countries. The decision to conduct a rapid review was a pragmatic decision due to time and resource limitations. Moreover, the streamlined methods of rapid reviews are particularly advantageous, as they can provide a timely synthesis of evidence that can be used to inform policy- and decisionmakers.

The research team also carried out comprehensive searches of professional body and charity websites, identifying several other publicly available self-management toolkits (e.g., Canadian-based Job Demands and Accommodation Planning Tool, https://aced.iwh.on.ca/jdapt/worker-en/access, accessed on 1 March 2024; UK Mind’s Wellness Action Plan, https://www.mind.org.uk/media/lbahso3x/mind-wellness-action-plan-workplace.pdf, accessed on 1 March 2024). However, evaluations of these toolkits are yet to be published, and they are not considered to offer directed guidance in a simple and practical solution-based format for all stakeholders to use together. Thus, our review offers an up-to-date synthesis of the evidence on self-management promotion for workers with long-term health conditions and disabilities, providing a strong foundation for further research in this area.

## 5. Conclusions

The current rapid systematic review highlights the requirement for workplace-based self-management interventions that are resource-efficient and address the needs of both workers and employer stakeholders. It also emphasises the importance of further studies that aim to understand the impact and implementation of interventions in practice, which may be crucial for their success. This review is timely in synthesising the current literature on workplace interventions for the self-management of long-term health conditions and disabilities, which could inform future research and policies in the work–health domain. The inclusion of high-quality studies enhanced the value of this review. Nevertheless, due to the limited number of studies and heterogeneity, further high-quality evidence that consistently measures the same outcomes should be prioritised in this field.

## Figures and Tables

**Figure 1 ijerph-21-01714-f001:**
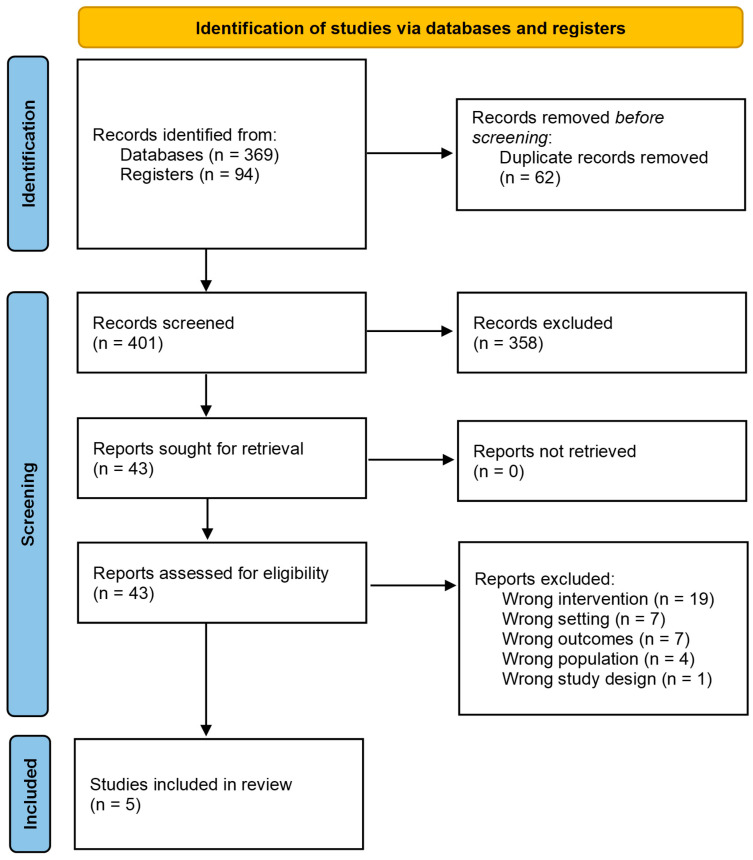
Selection of studies for the rapid review based on the Preferred Reporting Items for Systematic Reviews and Meta-Analyses (PRISMA) flow diagram.

**Table 1 ijerph-21-01714-t001:** Inclusion and exclusion criteria specified in terms of the participants/population, intervention(s), comparator(s)/control, outcomes, and study designs (PICOS).

PICOS	Inclusion	Exclusion
Participants/ population	Workers with any long-term health condition/disability that requires ongoing management for >12 months. Workers of any age (≥18 years), gender, nationality, ethnicity, seniority, occupation (e.g., manual/office based, low/high skilled, low/high income), or employment status (e.g., full/part time, permanent, contracted/subcontracted, volunteer).	Workers with health conditions that do not require ongoing management for >12 months (e.g., acute injury). Studies involving populations that do not focus exclusively on workers (e.g., patients in healthcare settings, unemployed etc.).
Intervention(s)	Any intervention that aims to promote self-management in workers with any long-term health condition/disability. Interventions that target multiple long-term health conditions/disabilities. Any mode of delivery (e.g., in-person, remote, digital), frequency, or duration.	Interventions not focused on self-management in workers. Interventions that address a single long-term health condition/disability.
Comparator(s)/control	Passive (e.g., no intervention, standard/usual care) or active (e.g., another type of self-management support).	No comparator specified.
Outcomes	Work productivity, work engagement, self-management, work ability, quality of life, psychological wellbeing, workplace fatigue, job satisfaction, work-based attendance, work self-efficacy, and measures of condition-specific health-status.	None of the pre-specified outcome domains measured, as outlined in inclusion.
Study designs	Quantitative-only studies or mixed-methods studies that report quantitative data, including randomised or non-randomised controlled trials and before and after studies.	Qualitative-only studies (e.g., focus groups, semi-structured interviews), expert opinions, practice or procedure guidelines, case reports, case series, abstracts from conferences, book chapters.

**Table 2 ijerph-21-01714-t002:** Review authors’ judgement using the Cochrane Risk of Bias 2 (RoB2) tool to assess the risk of bias of each included study for the following: A. randomisation process, B. deviations from intended interventions, C. missing outcome data, D. measurement of the outcome, and E. selection of the reported result.

Study	A	B	C	D	E	Overall
Haynes et al. [44]	Low	Low	Low	Low	Some concerns	Low
Schopp et al. [47]	Low	Low	Low	Low	Low	Low
Shaw et al. [45]	Low	Low	Low	Low	Low	Low
Smith et al. [46]	Low	Low	Low	Low	Low	Low
Wilson et al. [48]	Low	Low	Low	Low	Some concerns	Low

## Data Availability

Data are available on request from the first author.

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
