# Peer review of "A Rapid Systematic Review Assessing the Effectiveness of Interventions to Promote Self-Management in Workers with Long-Term Health Conditions and Disabilities"

_ijerph, 2024, doi:10.3390/ijerph21121714_

Round 1
Reviewer 1 Report
Comments and Suggestions for Authors― The limited number of included studies (five) and the focus on English-language publications restrict the area of applicability of the review. While this is a common limitation in rapid reviews, it should be explicitly noted in the editorial process.
― Although the introduction references several systematic reviews and interventions, it does not thoroughly examine why existing healthcare-based self-management interventions are challenging to adapt effectively to workplace contexts. Maybe this could improve the justification for the study's originality.
― The section briefly mentions that the rapid review was chosen due to time and resource constraints (lines 104–105). You could explore the possibility of adding a detailed justification for using a rapid review approach over a conventional systematic review, emphasizing how it meets the needs of stakeholders or addresses time-sensitive gaps in policy or practice.
― The databases searched, such as MEDLINE and APA PsycINFO (lines 183–185), focus on well-established sources. While grey literature databases were included, the section does not mention exploring non-English studies or data from low- and middle-income countries. It might be helpful if the authors could describe how grey literature findings were critically appraised, ensuring that they met a baseline quality standard. For example, they might explain whether these studies were weighted differently during synthesis due to their potential methodological limitations.
― The discussion highlights inconsistencies in the results (Lines 331–339) but does not explore the underlying factors in sufficient depth. Key elements such as intervention delivery methods, workplace culture, and participant characteristics are overlooked, which limits the ability to interpret why certain interventions were more effective than others. Addressing these factors could provide deeper insights into the variability observed in the outcomes. For example, differences in group-based versus individual interventions, or the influence of organizational support, might explain these discrepancies. To strengthen the discussion, consider citing specific examples from the included studies that illustrate how these factors impacted intervention effectiveness.
― To improve the Results section, it would be helpful to expand on outcomes such as workplace fatigue (Lines 291–293) and job satisfaction (Lines 295–299) to provide a more comprehensive analysis of the interventions’ impacts. Currently, these outcomes are only briefly mentioned, lacking sufficient interpretation or discussion of their relevance. Even if the findings for these outcomes are non-significant, they could offer relevant information about the limitations or challenges of workplace-based self-management interventions. It might be helpful to consider how these outcomes relate to employee productivity and wellbeing. For instance, if workplace fatigue did not significantly improve, discuss possible reasons, such as intervention duration or design limitations. Similarly, for job satisfaction, explore whether organizational factors like managerial support or job roles influenced the results. Including this level of detail will provide a fuller understanding of the interventions' effectiveness and highlight areas for further improvement or research.
― The need for future research is mentioned (Lines 340–343) but could be expanded to include specific areas such as digital interventions or tailored approaches for different workplace settings.
Author Response
Comment 1: The limited number of included studies (five) and the focus on English-language publications restrict the area of applicability of the review. While this is a common limitation in rapid reviews, it should be explicitly noted in the editorial process.
- Response 1: We acknowledge that the limited number of included studies and the focus on English-language publications restrict the scope and applicability of the review. In response to your feedback, we have explicitly noted these limitations in the revised manuscript, stating that restricting to English-language publications could introduce potential language or cultural biases, limiting the generalizability of the review findings (page 11, lines 436-438).
Comment 2: Although the introduction references several systematic reviews and interventions, it does not thoroughly examine why existing healthcare-based self-management interventions are challenging to adapt effectively to workplace contexts. Maybe this could improve the justification for the study's originality.
- Response 2: Thank you for this valuable suggestion. We recognize that interventions designed for healthcare settings often face significant challenges when adapted to workplace contexts. This may be due to various factors, such as employers lacking the necessary infrastructure (e.g., training to enhance manager knowledge and awareness) or self-management not being prioritized as an organizational objective. In response to your feedback, we have revised the manuscript to include these considerations, which we feel strengthens the justification for the study's originality (page 2, lines 85-89).
Comment 3: The section briefly mentions that the rapid review was chosen due to time and resource constraints (lines 104–105). You could explore the possibility of adding a detailed justification for using a rapid review approach over a conventional systematic review, emphasizing how it meets the needs of stakeholders or addresses time-sensitive gaps in policy or practice.
- Response 3: Thank you for raising this important point. We agree that rapid reviews offer significant advantages, particularly in addressing the “time-sensitive” needs of key stakeholders, such as employers, managers, and workers, by facilitating the timely translation of research evidence into practice. In this context, a rapid review is particularly relevant as the promotion of self-management for long-term conditions and disabilities in the workplace is becoming an increasingly urgent priority for public health policymakers, health services, and professional organizations. To address your suggestion, we have revised the manuscript to emphasize these considerations and to further strengthen the justification for using this approach (page 3, lines 109-114 and 119-122).
Comment 4: The databases searched, such as MEDLINE and APA PsycINFO (lines 183–185), focus on well-established sources. While grey literature databases were included, the section does not mention exploring non-English studies or data from low- and middle-income countries.
- Response 4: Thank you for your comment. In line with best practice recommendations from the Cochrane Rapid Reviews Methods Group, our searches were restricted to English-language publications. In our original manuscript, we recognized this as a potential limitation, and this point has been further elaborated. Regarding data from low- and middle-income countries, no studies meeting the eligibility criteria were identified that examined this context. This may also be influenced by the restriction to English-language publications, which we have also acknowledged as an additional limitation (page 11, lines 436-438).
Comment 5: It might be helpful if the authors could describe how grey literature findings were critically appraised, ensuring that they met a baseline quality standard. For example, they might explain whether these studies were weighted differently during synthesis due to their potential methodological limitations.
- Response 5: Thank you for your comment. Two grey literature sources (Social Care Online from the Social Care Institute of Excellence and ProQuest Dissertation & Theses Global) were included to enhance comprehensiveness and mitigate potential publication bias. To assess quality of non-randomized studies, including grey literature, we planned to employ the Risk Of Bias In Non-randomized Studies of Interventions (ROBINS-I) tool. This was outlined in our prospectively registered protocol in the International Prospective Register of Systematic Reviews (PROSPERO, registration number: CRD42023475510). This information has now been added to the revised manuscript for transparency (page 6, line 231-233). However, as all included studies were randomized controlled trials published in peer-reviewed journals, the ROBINS-I tool was not required or reported in this review. This clarification has also been incorporated into the revised manuscript to enhance the clarity of our methodological approach (page 8, lines 323-324).
Comment 6:
The discussion highlights inconsistencies in the results (Lines 331–339) but does not explore the underlying factors in sufficient depth. Key elements such as intervention delivery methods, workplace culture, and participant characteristics are overlooked, which limits the ability to interpret why certain interventions were more effective than others. Addressing these factors could provide deeper insights into the variability observed in the outcomes. For example, differences in group-based versus individual interventions, or the influence of organizational support, might explain these discrepancies. To strengthen the discussion, consider citing specific examples from the included studies that illustrate how these factors impacted intervention effectiveness.
- Response 6: Thank you for highlighting these important considerations. Across all included studies, the interventions were consistent in delivery mode and duration. Specifically, they involved group-based interventions delivered in person by trained facilitators, typically comprising one 2.5-hour session per week over a six-week period. However, whether alternative formats (e.g., individual-based approaches, digital or remote delivery, or variations in session length) would yield comparable outcomes remains an area for future investigation. Additionally, the studies assessed interventions in diverse organizational settings (private, public, and third/voluntary sectors) and included participants with varying long-term conditions, disabilities, job roles, and levels of seniority. These organizational and participant demographic or occupational variables likely contributed to the variability in outcomes but were not fully explored in the included studies. Further research is needed to examine the influence of these factors in greater depth, which could provide valuable insights into how they impact intervention effectiveness. To address your feedback, we have expanded the Discussion section to incorporate these points (page 9, lines 356-368).
Comment 7: To improve the Results section, it would be helpful to expand on outcomes such as workplace fatigue (Lines 291–293) and job satisfaction (Lines 295–299) to provide a more comprehensive analysis of the interventions’ impacts. Currently, these outcomes are only briefly mentioned, lacking sufficient interpretation or discussion of their relevance. Even if the findings for these outcomes are non-significant, they could offer relevant information about the limitations or challenges of workplace-based self-management interventions. It might be helpful to consider how these outcomes relate to employee productivity and wellbeing. For instance, if workplace fatigue did not significantly improve, discuss possible reasons, such as intervention duration or design limitations. Similarly, for job satisfaction, explore whether organizational factors like managerial support or job roles influenced the results. Including this level of detail will provide a fuller understanding of the interventions' effectiveness and highlight areas for further improvement or research.
- Response 7: Thank you for your insightful comment. We agree that several factors may explain the inconsistent findings reported between studies. First, the studies employed different measures to assess each outcome domain, which likely contributed to variability in the results. Second, limited information was provided in some studies in terms of participant characteristics, including the specific long-term conditions or disabilities experienced. Third, certain outcomes, such as workplace fatigue or job satisfaction, may not have been directly targeted by the interventions. These outcomes may be beyond the control of individual workers and instead depend on changes to organizational factors, such as such as workplace culture, managerial support, and punitive policies. While these outcomes were only briefly mentioned in the Results section, we felt that a deeper exploration of these factors was more appropriately suited to the Discussion (page 9, lines 345-355).
Comment 8: The need for future research is mentioned (Lines 340–343) but could be expanded to include specific areas such as digital interventions or tailored approaches for different workplace settings.
- Response 8: Thank you for this valuable suggestion. In response, we have expanded the Discussion section to emphasize the need for future research that explores the development and evaluation of interventions delivered through alternative formats, such as digital or remote approaches. Additionally, we have highlighted the importance of tailoring interventions to suit different workplace settings, including variations in organizational size, type, and sector. These additions have been incorporated into the manuscript to provide a more comprehensive discussion of priorities for future research (page 10, lines 399-401).
Reviewer 2 Report
Comments and Suggestions for Authors
1. Expand the search of other scientific databases, especially since the number of publications (records) received is small.
2. Present the criteria for inclusion and exclusion of publications in the form of a table.
3. The text lacks a description (characteristics) of the reasons for exclusion of publications. The description in Figure 1 is not enough.
4. There is an error in paragraphs: 80,120,261. It should be diabetes (type I and II).
5. Clearly state what scientific gap this review fills.
Author Response
Comment 1: Expand the search of other scientific databases, especially since the number of publications (records) received is small.
- Response 1: Thank you for your suggestion. In accordance with best practice recommendations for conducting rapid reviews outlined by the Cochrane Rapid Reviews Methods Group, it is advised that a focused number of information sources (at least two) be selected to efficiently retrieve relevant literature. Based on this guidance, we made the pragmatic decision to search three well-established databases: APA PsycINFO, APA PsycARTICLES, and MEDLINE. Together, these databases encompass approximately 175 million references to peer-reviewed journal articles across the behavioural, social, and life sciences, ensuring coverage of the most relevant and high-quality research in the field. To enhance comprehensiveness and reduce potential publication bias, we also included two grey literature sources: Social Care Online and ProQuest Dissertations & Theses Global (PQDT Global). Social Care Online is one of the UK’s largest databases for research, reports, policy documents, and journal articles in social care and social work, while PQDT Global is the world’s most extensive collection of full-text dissertations and theses, with over 5 million citations and 3 million full-text works from thousands of universities worldwide. These additional sources were chosen to capture unpublished or non-peer-reviewed material that may still provide valuable insights.
Comment 2: Present the criteria for inclusion and exclusion of publications in the form of a table.
- Response 2: Thank you for your suggestion. In response, we have added a new table (Table 1) to the manuscript, which presents the inclusion and exclusion criteria in a structured format for greater clarity and ease of reference. This addition can be found on page 5.
Comment 3: The text lacks a description (characteristics) of the reasons for exclusion of publications. The description in Figure 1 is not enough.
- Response 3: Thank you for highlighting this important point. To address your feedback, we have expanded the text to include additional details on the reasons for excluding publications, along with specific examples for clarity. This information has been added to the manuscript to complement the description provided in Figure 1 (page 6, lines 253-258).
Comment 4: There is an error in paragraphs: 80,120,261. It should be diabetes (type I and II).
- Response 4: This error has been corrected to “diabetes (type I and II)” throughout the revised manuscript.
Comment 5: Clearly state what scientific gap this review fills.
- Response 5: Thank you for your comment. The current review addresses a significant scientific gap as it is the first to systematically synthesize evidence on the effectiveness of workplace-based interventions aimed at promoting self-management for individuals with any long-term condition or disability. This synthesis not only highlights the current state of evidence but also identifies areas where further research is needed, offering valuable insights to inform the design, evaluation, and implementation of more effective workplace interventions. These points have been explicitly added at the beginning of the Discussion section to clearly articulate the unique contribution of this review (page 9, line 336-338 and 342-344).
Reviewer 3 Report
Comments and Suggestions for Authors
Dear authors, first of all, I would like to congratulate you on the study carried out.
I would like to propose a number of suggestions that need to be implemented to improve your work:
- In the abstract, at the beginning, add the word objective.
- In Figure 1, the citations should be in Vancouver style.
- Section 4.2. Also add the chosen databases as limitations. It is possible that studies published in other electronic resources may have been unintentionally omitted.
Author Response
Comment 1: I would like to congratulate you on the study carried out.
- Response 1: Thank you for your positive feedback.
Comment 2: In the abstract, at the beginning, add the word objective.
- Response 2: The abstract has been revised accordingly to address your comment.
Comment 3: In Figure 1, the citations should be in Vancouver style.
- Response 3: Thank you for bringing this to our attention. No citations were provided in Figure 1. However, the citations provided in the risk of bias table (originally Table 1, now Table 2) were not in Vancouver style. These have now been corrected.
Comment 4: Section 4.2. Also add the chosen databases as limitations. It is possible that studies published in other electronic resources may have been unintentionally omitted.
- Response 4: Thank you for highlighting this point. While our database selection aligns with best practice recommendations for conducting rapid reviews, we acknowledge the possibility that relevant publications within the scope of our review may have been unintentionally omitted due to the limited number of information sources selected (i.e., APA PsycINFO, APA PsycARTICLES, MEDLINE, Social Care Online from the Social Care Institute of Excellence, ProQuest Dissertation & Theses Global). We have now added this limitation to the manuscript (page 11, lines 429-436).
Reviewer 4 Report
Comments and Suggestions for Authors
Dear authors
The manuscript entitled “A rapid systematic review assessing the effectiveness of interventions to promote self-management in workers with long-term health conditions and disabilities.” refers to a meritorious work, the subject being pertinent, methodologically correct, following the PRISMA protocol. The text is clear and objective, and the doubts that arose during reading the manuscript were generally resolved in points 4.1 and 4.2 of the discussion.
On page 7, the topic “3.1.3. Interventions”, is repeated on line 270. I suggest that it be removed.
Author Response
Comment 1: The manuscript entitled “A rapid systematic review assessing the effectiveness of interventions to promote self-management in workers with long-term health conditions and disabilities.” refers to a meritorious work, the subject being pertinent, methodologically correct, following the PRISMA protocol. The text is clear and objective, and the doubts that arose during reading the manuscript were generally resolved in points 4.1 and 4.2 of the discussion.
- Response 1: Thank you for your kind feedback.
Comment 2: On page 7, the topic “3.1.3. Interventions”, is repeated on line 270. I suggest that it be removed.
- Response 2: Thank you for highlighting this typographical error. This has now been amended to the correct subheading, “3.1.4. Effectiveness” (page 8, line 281).
Reviewer 5 Report
Comments and Suggestions for Authors
Topic is very interesting and less presented in existing literature. The paper is well organized and documented.
Observations:
2.1.4. Outcomes. It is not clear why you have included these outcomes in your analysis and not others? (eg work productivity, work engagement, self-management …). Is there any risk expected these outcomes" to be highly correlated with each other?
3.1.3. Interventions – explanation for the “k” (k = 3, where k means …)
Table S3: In there any risk of bias in assuming the “statistically significant positive intervention effect” in Table S3 taken into consideration that the investigated studies used different research methodologies (samples, environmental variables, scales …). If so, a short explanation should be given either as a limitation of the study or as a “low risk"
Author Response
Comment 1: Topic is very interesting and less presented in existing literature. The paper is well organized and documented.
- Response 1: Thank you for your encouraging feedback.
Comment 2: 2.1.4. Outcomes. It is not clear why you have included these outcomes in your analysis and not others? (eg work productivity, work engagement, self-management …). Is there any risk expected these outcomes" to be highly correlated with each other?
- Response 2: Outcomes were selected in consultation with the wider research team, our project-specific PPIE group, and a recently published core outcome set for workplace-based interventions. This information has now been included in the revised manuscript (page 4, lines 173-175).
Comment 3: 3.1.3. Interventions – explanation for the “k” (k = 3, where k means …)
- Response 3: Thank you for your comment. In systematic reviews, “k” can be used to indicate the number of included studies that measured a specific outcome. However, we have decided to remove this notation from the manuscript, as the same information is already provided in the text, to avoid unnecessary repetition.
Comment 4: Table S3: In there any risk of bias in assuming the “statistically significant positive intervention effect” in Table S3 taken into consideration that the investigated studies used different research methodologies (samples, environmental variables, scales …). If so, a short explanation should be given either as a limitation of the study or as a “low risk".
- Response 4: Thank you for highlighting this important point. We acknowledge that the differing methodologies employed across studies may have influenced the review findings and introduced variability. First, the studies employed different measures to assess each outcome domain, which likely contributed to variability in the results. Second, limited information was provided in some studies in terms of participant characteristics, including the specific long-term conditions or disabilities experienced. Third, certain outcomes, such as workplace fatigue or job satisfaction, may not have been directly targeted by the interventions. These outcomes may be beyond the control of individual workers and instead depend on changes to organizational factors, such as such as workplace culture, managerial support, and punitive policies. These points have now been added to the manuscript (page 9, lines 345-355).
Reviewer 6 Report
Comments and Suggestions for Authors
Dear authors,
Thank you for your submission.
This is a well-written and informative paper.
I have only few review comments, suggestions, and recommendations for your consideration as follows:
Introduction: for better readability, please start by providing definitions of long-term health conditions and disabilities.
Eligibility criteria: please add exclusion criteria as well (if any) and be more specific for participants/population, intervention(s), comparator(s)/control, outcomes, search strategy, and study selection.
2.1.2. Intervention(s): can you elaborate more on selected interventions? were there any specifications in terms of intensity of selected intervention (frequency and duration)? who provided these interventions?, etc.
2.1.4. Outcomes: can you add your rationale of why those outcomes were selected for this study? and for pathological or clinical assessment studies selected, were those assessments (outcome measures) valid and reliable? please elaborate.
Figure 1: what do you mean by wrong intervention, setting, outcomes, population, study design? maybe you can add a caption under figure for clarification.
Please address the mentioned above and will be happy to look at your revised manuscript.
Many thanks.
Best wishes,
Author Response
Comment 1: This is a well-written and informative paper.
- Response 1: Thank you for your positive feedback.
Comment 2: Introduction: for better readability, please start by providing definitions of long-term health conditions and disabilities.
- Response 2: Thank you for this suggestion. A suitable definition has been provided at the start of the Introduction (page 9, lines 34-36).
Comment 3: Eligibility criteria: please add exclusion criteria as well (if any) and be more specific for participants/population, intervention(s), comparator(s)/control, outcomes, search strategy, and study selection.
- Response 3: Thank you for your suggestion. In response, we have added a new table (Table 1) to the manuscript, which presents the inclusion and exclusion criteria in a structured format for greater clarity and ease of reference. This addition can be found on page 5.
Comment 4: 2.1.2. Intervention(s): can you elaborate more on selected interventions? were there any specifications in terms of intensity of selected intervention (frequency and duration)? who provided these interventions?, etc.
- Response 4: Further information regarding interventions is now provide in a new table (Table 1). Specifically, any workplace-based intervention could be included that aimed to promote self-management in individuals with any long-term health condition/disability. Interventions should target multiple long-term health condition/disability, and any mode of delivery (e.g., in-person, remote, digital), frequency, or duration was eligible for inclusion.
Comment 5: 2.1.4. Outcomes: can you add your rationale of why those outcomes were selected for this study? and for pathological or clinical assessment studies selected, were those assessments (outcome measures) valid and reliable? please elaborate.
- Response 5: Outcomes were selected in consultation with the wider research team, our project-specific PPIE group, and a recently published core outcome set for workplace-based interventions. This information has now been included in the revised manuscript (page 4, lines 173-175).
Comment 6: Figure 1: what do you mean by wrong intervention, setting, outcomes, population, study design? maybe you can add a caption under figure for clarification.
- Response 6: Thank you for highlighting this important point. To address your feedback, we have expanded the text to include additional details on the reasons for excluding publications, along with specific examples for clarity. This information has been added to the manuscript to complement the description provided in Figure 1 (page 6, lines 253-258).